# Creativity as a Key Constituent for Smart Specialization Strategies (S3), What Is in It for Peripheral Regions? Co-creating Sustainable and Resilient Tourism with Cultural and Creative Industries

**Christopher Meyer** [1,2,]*[iD], **Laima Gerlitz** [2][iD] **and Monika Klein** [3]

1   Department of Business Administration, School of Business and Governance, Tallinn University of Technology, Ehitajate tee 5, 19086 Tallinn, Estonia
2   Wismar Business School, Hochschule Wismar, University of Applied Sciences: Technology, Business and Design, Philipp-Müller-Str. 14, 23966 Wismar, Germany; laima.gerlitz@hs-wismar.de
3   Finance and Management, Faculty of Economics, University of Szczecin, 70-453 Szczecin, Poland; monika.tomczyk@usz.edu.pl
*   Correspondence: chmeye@taltech.ee or christopher.meyer@hs-wismar.de

**Abstract:** Sustainable tourism is one of the key sectors in the South Baltic Sea Region (SBSR), which belongs to the role model for sustainability—the Baltic Sea Region (BSR). In this context, resilience, recovery and sustainability become key common threads calling for new approaches mitigating negative impacts, upscaling resilience capacity and boosting recovery in the post-pandemic era. The present work aims at revealing conceptual and practical pathways for policy makers and businesses in revitalizing sustainable tourism in the region by emphasizing cultural and creative industries (CCIs) as strong contributors to sustainable development and economic ecosystems, such as tourism. Tourism is also one of the key thematic areas of the smart specialization strategies (S3) in the SBSR. However, there is almost no link between CCIs' potential for sustainable and resilient tourism and their contribution to the co-design and co-creation of S3. CCIs are rather absent agents in quadruple helix networks supporting S3 policy implementation. The literature on this topic is still premature, and represents a clear gap in knowledge. By virtue of these circumstances, the present research investigates how CCIs contribute and reveal new linkages between local assets, potential markets and societal challenges by engaging them as proven sustainable innovation and transition brokers in transnational quadruple helix partnerships following S3 policies in accordance with the sustainable development goals (SDGs), thus supporting sustainable and resilient tourism. Moreover, this paper aims at advocating for development of rural and peripheral regions, thus reducing the so-called "rural marginalization". In addition, this paper also supports ongoing recent discussions on related vs. unrelated diversification policy within the S3 realm.

**Keywords:** CCIs; smart specialization strategies (S3); regional innovation strategies on smart specialization (RIS3); sustainability; resilience; South Baltic Sea Region; peripheral region; quadruple helix; transformative innovation policy; ecosystems



## 1. Introduction

The present research contribution aims at revealing the increasing role of creativity and cultural and creative industries (CCIs) in regional policy design and implementation, i.e., smart specialization strategies (S3) within regional innovation strategies (RIS) and regional diversification policies. The literature linking CCIs and S3 is premature. After the popular research treatise of Cooke and De Propris (2011) positioning the role of CCIs for the EU's smart growth and linking it with place-based specialization debates [1], the following 10 years barely contributed to this research pool. Indeed, most research, in addition to practical work, in response to EU regionalization and cohesion policy implementation

in the context of CCIs, avoids twinning S3 and cultural heritage [2–4]. This constitutes a clear gap in the knowledge and serves as an essential impetus for the authors to envisage the role of CCIs in the S3 policy nexus, especially considering the fact that S3 stands for strategic broader innovation orientation and place-based approaches to tackle existing innovation challenges on the one hand, and CCIs being acknowledged for their strategic partner role in innovation development on the other. As a result, this research does not call for providing support to CCIs via policies [5–7], but rather for the endorsement of regional policies through CCI interventions. The authors argue that CCIs' role is much more than that of strengthening cultural heritage integration in S3 policy design.

Screening available project depositories, CCIs' exploitation for regional development has focused on either strengthening a single industry (e.g., CCIs' internationalization, CCIs' innovation capacity and digitalization) or cross-industry innovation (e.g., boosting social or digital innovation capacity in other industries through CCIs). As a result, a large number of projects related to CCIs and cross(industry)-innovation revealed the benefits of multilateral approaches and yielded positive effects on innovation through networks of relationships and alliances [8–10]. However, within this context of either formal or informal collaboration modes, the focus has largely been laid on innovation performance and its outputs. Any projects on CCIs' potential for regional policy design and implementation, e.g., through RIS and S3, are so far scant. Only 10 percent of the 243 S3 strategies give priority to culture [4] (p. 12). In sum, there are almost no projects on the exploitation of CCIs and their strategic partnerships in diverse collaboration models, e.g., quadruple helices, and how helix actors and their collaboration governance patterns shape innovation for civil society. In such a regional policy setting, the exploration of CCIs' role as innovation brokers (mediators, collectors and connectors) is rather absent, and thus this urges the present work.

Against this background, the paper aims at strengthening CCIs' role for innovation policy design and implementation through S3 in rural, remote and peripheral regions, thus reducing the so-called "rural marginalization" [11], which is to be understood as not only geographical, but rather as relational remoteness, characterized by broken or loosening socioeconomic and political connections and interactions, such as industrial decline, an ageing population, low education levels, land abandonment, and unemployment, which are embedded in the process of social change (p. 556). Indeed, the intensifying race for specialization and competition among regions has led to the prioritization of urban and metropolitan development over rural development, and has highlighted growth of industrial regions in the EU through S3. This, in turn, has generated disproportions in the consideration of urban and rural development through policy instruments, while also affecting CCI-related discourse [12–15].

In this light, the research in hand also supports the emerging body of literature on spatial issues of innovation, by positioning CCIs' role for policy development, and its endorsement in particular, in the periphery [16–20], overcoming the isolation and marginalization of locations at the periphery of development, questioning the significance of relationships and interactions in the given environment, e.g., ecosystems, followed by nurturing people-centered approaches to development, strengthening endogenous potential capitalization and local participation and facilitating the rethinking of local resources [21] (p. 159). In this sense, the present paper aims at expanding the existing literature on successful innovative regions by also exploring the benefits that peripheral regions might have and/or develop in innovation modes through CCIs' intervention, thus overcoming poor governance and lack of financing [22] (p. 137). In the light of these pivotal challenges, the potential of CCIs in peripheral regional development and policy support becomes essential in the given study region of the SBSR, in particular by reflecting upon strongly pursued related diversification policy in the frame of S3 [23,24] in the tourism sector. As a result of focusing on this related specialization, the SBSR becomes vulnerable to disruptions, e.g., caused by COVID-19, or, recently, the rapid pace of digital and environmental regulation-driven twin transition. Despite the huge potential associated with the environmental and cultural assets available, the region has failed so far in building

up sustainable and innovative approaches and collaborative models and unleashing the benefits of the peripheral localities [25] (p. 7).

Given both the advantages and challenges linked with CCIs, their potential is associated with collaborative modes and strategic partnerships, in which different actors engage in formal and informal collective decision-making processes aiming at public policy improvement. This exactly applies to S3 as a public innovation policy. Thus, reinforcing CCIs' role through S3 strategies in peripheral and marginalized regions, there is no doubt that they might spur wider economic development, with a huge record of evidence [26–30]. CCIs play a important role in both the urban economy [31–33] and regional or local development [34–37]. Indeed, there is no doubt that CCIs act as drivers and accelerators of the economy at all geographical scales [38–40]. What remain highly disputed are the reciprocal impacts of CCIs on urban and rural development, as they can increase inequalities between places, favoring on average the most developed places [38] (p. 16). Similar inequalities related to CCIs' impacts are also expressed through sustainability performance, as cities with high innovation performance and a strong density of CCIs have the worst social and economic inequality and increased pollution, resource scarcity or paucity of housing. This is because CCIs employ highly skilled workers, leading to a rise in wages and social gaps [41]. By contrast, a lower presence of CCIs in remote regions results in higher vulnerability, low networking and low level of institutional support. Such regions record higher unemployment and higher competition. In this light, within the rationale of this present work to combine CCIs with S3 strategies in the given regional setting by lending more importance to collaboration modes, networks of interrelationships become pivotal, since local conditions, e.g., the environment and interactions of CCIs in the given ecosystems, can differently affect the concerned localities [38] (p. 16). Similarly, since an ecosystem concept integrates operational, environmental, economic, technological, social and legal dimensions [42–44], as well as implying the causality and interdependencies of assets, institutions, knowledge and human capital [45–47], the exploration of social and institutional aspects for sustainable and resilient co-creation through social and institutional interactions and networks of regional development paths is inevitable.

In the face of rising inequality, such environmental, social and economic challenges appear to multiply and push policy makers and all affected actors to the hilt. Indeed, these conflicting interests frame today's business environment and the so-called triple bottom line (TBL) as an interplay of environmental, social and economic components, highlighting the significance of harmonizing business sustainability efforts in these three elements dimensions [48–50]. Extreme disturbances of supply and value chains as well as disruption of economic and social ecosystems due to the COVID-19 pandemic bring about constant fluctuations and create turbulent times. In this context, resilience, recovery and sustainability become key common threads not only in policy discourse, but also in the academic realm, calling for new approaches for mitigating negative impacts, upscaling resilience capacity and boosting recovery [51–54]. Indeed, CCIs could act here as an expedient to instigate transition, aiming at (re)building resilient capacity and finding novel ways to break through the lockdown [55–57], to face personal and professional adversities [58], and take advantage of a gamut of micro-resilience aspects, such as flexible and adaptive environments and creative and human capital [59]. Available diversified networks of CCIs are able to enhance innovation and capitalization through synergies and linkages with economic ecosystems, once supported by public stimulus. This serves as a precondition for the sustainable development of regions [60] (p. 1).

In order to reveal sustainable and resilient pathways for regions based on the case of tourism sector, the paper in hand raises the research question of *how CCIs, already acknowledged as knowledge and innovation brokers, can support and endorse S3 strategies' design and implementation, given the case of tourism sector? How do they engage in and shape Quadruple helix partnerships that are essential enablers of S3?* The investigation starts with building up the interrelated conceptual fundamentals, linking CCIs with sustainability and S3, followed by positioning this intertwining in the theoretical ream. Recalling recent appeals for open

and evidence-based research designs in both the CCIs and innovation policy discourses, the work deploys empirical data from the EU Interreg project "CTCC—Creative Traditional Companies Cooperation" which was implemented under the Interreg South Baltic Program 2014–2020 from July 2017 to December 2021. By implementing innovation prototypes (product, service, marketing and business model development) in the tourism sector in individual regions across the SBSR, the work counts on strong practice-based empirical evidence, which, when analyzed, evaluated and synthesized, enables us to conceptualize and fertilize the co-creation and reinforcement of S3 strategies in the upcoming EU and regional development cohesion period, 2021–2027. By virtue of the practical nature of this paper, the lessons learnt and managerial implications can support policy makers and regional planners in revamping S3-related discourses, principally by bringing in CCIs as strong contributors, accelerators and enablers into recent innovation and regional development policy discourses, in particular in peripheral regions.

## 2. Linking Up, Intertwining and Synthesizing Conceptual Foundations

Little is known about CCIs' potential for sustainable regional development and their interplay with regional innovation policies, such as S3. In order to underpin this research focus based on the interplay of CCIs, sustainability and S3 policy frameworks in a given regional setting, the authors of this study encapsulate them by reflecting their interlinkage in the available body of literature based on the common ground on the one hand, and by tracing their conceptual vestiges, thus linking up with key theoretical treatises, on the other.

### 2.1. What Is Common among CCIs, Sustainability and Smart Specialization Strategies (S3)?

Bearing in mind the interplay of environmental, social and economic dimensions in business sustainability efforts, creativity clearly fits into this discourse. Creativity is seen as a strong partner and part of sustainability transition. Creativity can be perceived as the heart of sustainability, rooted in sustainable social, economic, environmental and cultural practices, as well as an enabler and driver of development [61] (pp. 65–66). Considering this, CCIs have been emphasized in global, EU and regional policy discourse, in particular supporting sustainable development in the frame of the Agenda 2030 of the United Nations (UN), aiming at revealing the economic value of CCIs since 2008 [62]. Core strengths of CCIs are also associated with their ability to broker and share knowledge, craft innovative policies, and produce as well as mobilize digital tools [61] (pp. 68–69). Scholars in Europe have also raised the necessity to integrate CCIs into policy discourses and regional development paradigm, by aligning policies of the EU Member States with the UN SDGs [63], including monitoring the contribution of culture to the SDGs and engaging CCIs as a pillar for the sustainable development paradigm on all levels and across disciplines [64] (pp. 65–55), [65] (p. 23).

However, policy endeavors so far miss clear actions recognizing CCIs' value for the SDGs, which can be encountered through sectoral integration, a cross-sector and multi-sector industry partnership [66] (pp. 37–38). The missing link also shapes existing scholarly and practical contributions regarding CCIs for sustainability, which are still to a high extent limited and burgeoning [67–72]. Most of the existing entries avoid the role of culture, instead focusing on overall CCIs and their intervention in sustainable development [73,74]. Indeed, culture as a view of an artistic expression cannot equal the perception of CCIs [75] (p. 153). More endeavors are also needed to strengthen strategic paths of culture [76] and question the role of CCIs for sustainability rather than searching for vestiges of CCIs' sustainability [77,78], which can be traced back to endogenous conditions and resources [79].

Sustainability requires efficient and cross-sectoral partnerships and collaboration among multiple diverse partners [80]. Since CCIs are valued as being enablers and contributors to sustainable development, their role in cross-sectoral partnerships appears to be indispensable. Collaboration with CCIs endorses activities that are linked to social and cultural development of the locality [81] (p. 20). The role of institutional and social aspects

is also important when perceiving sustainability as a means of meeting the needs of today without compromising the ability of the future generations to meet their own needs [82] (p. 43). Indeed, it can be stressed that CCIs can affect and increase the level of resilience in socio-ecological systems, not only at the organizational level [83] (p. 1216). However, little is known about how cross-sectoral collaboration can create value for various collaborating stakeholders, thus enabling transition and overcoming sustainability problems, residing in tensions over social, environmental and economic dimensions [84] (p. 1039). One potential source is related to overcoming human alienation from nature, thus bringing the ecological imperative back to the forefront, harmonizing economic expansion and ecological limits [85] (p. 54), which leads to new social constructs embedded in the system of values, norms and culture.

This, in turn, is encapsulated within the innovation itself, which from the evolutionary perspective is seen as a complex systemic process integrating iterative (evolutionary) interactions from the Shumpeterian cycle perspective between newly produced knowledge, networked learning and institutional support, embedded within functional, geographic and social boundaries of a particular regional ecosystem, thus paving the way for regional adaptation and resilience [86] (p. 114). In this sense, referring to CCIs as having a strongly symbolic (art-based) nature of the knowledge base emerging from complex, dynamic and tacit interactions embedded within local settings might help to underpin regional resilience and boost the long-term ability for adaptation and entering into new development paths. For this reason, support policies, such as S3, should aim at creating interorganizational informal networks, fostering symbolic knowledge exchange and cooperation [87] (pp. 18–19). While developing sustainability strategies and key performance indicators, a more holistic approach is needed, e.g., by including social aspects ranging from tolerance to a climate for doing business, social capital and local leadership [88] (p. 20), cultural awareness, cross-sectoral competency and the structured inclusion of intermediaries, such as innovation mediators and catalysts [81] (p. 28), that, in turn, support local and regional authorities in managing S3 with capacity and resources [89] (p. 1386). This, however, still remains a challenge, since the EU has so far not made any pivotal strategic contributions towards implementing the 2030 Agenda of the UN in the cultural dimension, thus leaving a lot of room for maneuver to make culture an active contributor to sustainability [90] (pp. 86–87).

Acknowledging the strong and long-term influence of institutional support to innovative regions and regional resilience, sustainability has been a common thread and serves as a panacea to step out of the recession and depression caused by external shocks. Therefore, recent S3-related discussions have lent importance to sustainability aspects as well [91–98]. Sustainability will guide S3 policy development for 2021–2027 by refining S3 as a tool for innovative and smart economic transformation [91] (p. 4), building upon active stakeholder engagement, consultation and communication among multi-level actors within the policy design cycles [92] (pp. 38–45), as well as intensifying debates pertaining to the capacity of innovation policy to address complex sustainability problems and transformational change [93] (p. 3). Linking the new S3 policy direction with sustainability also allows one to boost the potential to provide more targeted and tailored support to enhance the recovery from the COVID-19 crisis as well as to accelerate twin (long-term and short-term) transitions [99] (p. 3).

In this context, S3, as a place-based approach, particularly supports economic and social recovery at the local and regional level, as a result of changing resilience and competition conditions [100] (p. 33). Rationally, this policy should be better embedded in lagging or peripheral regions [101–105]. Additionally, the potential of cross-border collaborations to achieve synergies should be further explored [106]. Paradoxically, such regions are facing higher challenges in capitalizing on S3 policy, mainly driven by missing excellence, which rather gives the floor to leading regions to capitalize on regional policies. As a response to this paradox, the revamping of S3 policy needs to focus more on the socio-ecological model of innovation in order to directly affect the quality of life, underpin experimental learning, design and implement business training and assistance models, encourage social

innovation and revitalize coalitions of regional and local actors, supported by intermediary institutions [101] (pp. 87–88). Here, CCIs are seen as crucial intermediaries. Indeed, lagging and peripheral regions show higher potential for social innovation governance, as they are composed of smaller communities that share closer social, political, integrative, cognitive and institutional proximity [107,108], thus becoming strong enough to break down rural marginalization [109] (pp. 59–60), overcoming local challenges through accumulated actions and processes [110] (p. 284), as well as reflecting burgeoning cross-sectoral and trans-local collaboration patterns, thus opening up new potentials and horizons [11] (p. 552). Indeed, research on the application of S3 at the bottom-up level of local governments, industries, clusters and enterprises remains scarce [111] (p. 1). For this task, overall, CCIs come into play as strong potential intermediaries able to break through the isolation lock-in boundaries.

*2.2. Conceptual Foundations Advocating CCIs in Sustainable Smart Specialization*

While research and policy entries on sustainable development within urban and regional nexuses are rising, CCIs' potential for the design and implementation of public policies, such as S3 in the discourse of RIS, is still stagnating. The screening of the literature yields only a few entries linking CCIs and S3, mainly through the lens of cultural heritage [112–115]. Sadly, the exploitation of creative potential, strategic partnerships and diverse collaboration models enabled by CCIs in the realm of S3 discussions, in particular pertaining to the new EU cohesion policy 2021–2027, is very scant. This represents a pivotal missing link, despite the already visible common traits that CCIs and S3 discourses share when focusing on sustainability (rf. Section 2.1). Sustainability discourses are common for regional innovation and regional studies, therefore also for S3.

Sustainability germinates from the search for new ways out of depressions and shocks caused by the COVID-19 pandemic. For this, actual place-based endogenous strengths and opportunities should be recognized, followed by increasing methodological improvements, which could build upon experimentation, self-discovery, and the inclusion of outsiders to diversify the knowledge base [116] (pp. 1679–1680). Fostering new partnerships between research organizations, enterprises and public authorities is a major concern of S3 strategies, calling for the setting up of new collaborative platforms. On a regional level, a European political answer to globalization, climate change and social exclusion is offered by S3 aiming at innovation-driven development, strengthening each region's competitive advantage, as well as increasing the system assets and the capability to learn [114] (pp. 8–9). Although there exists no direct linkage of CCIs with S3 in the sustainability nexus, practical policy papers have made the first attempts to bring in creative potential into the S3 policy design and implementation, mainly through concepts of design thinking [117], co-creation [118,119], self-discovery [120,121] and partnerships and networks [122]. As a result, by building on the analysis of the concepts and their interlinkages, the authors of this work contend that CCIs' role for S3 in the sustainability and place-based context can be traced back to the following common conceptual grounds: a) processual; b) institutional, and c) output-based.

Both CCIs and S3 are process-driven and output-oriented, seeking and improving innovation outputs and innovation capacity at the regional and local levels. Creativity stands at the core of CCIs, and therefore CCIs are essential enablers and sources of innovation, competitiveness and growth [123]. CCIs can promote manageability and improve understanding, change the entire process, implement new methods, influence strategies and thus affect the entire development process of a product or service [67] (p. 14). Creativity links design (form) and innovation. Indeed, by looping innovation discourse, creativity is a key ingredient for innovation, as coined by Schumpeter. It is a process of "creative destruction"—new ways using existing means, materials and methods. Something new can be created not from a regular basis, but rather from something that is new to the existing value system of a static economy. New is a new kind. It also involves using and/or employing something in a new manner, thus carrying out new combinations [124] (pp. 409–410). Innovation means creativity plus exploration [125] (p. 8). Indeed, in the

sustainability and policy mix context, the combination of old and new is essential [126] (p. 206), [127] (p. 195), giving floor to disruptive action and innovation and providing room for new market opportunities, through access to information, the identification of gaps between supply and demand and the commitment of scarce resources [128] (p. 35f), [129] (p. 287).

Creativity is an essential part of innovation, thus implying a growing need to determine methods, which could be used to generate more and better ideas, which then could be commercialized, thus turning into innovations [130] (p. 2). This links to an output-based common conceptual ground. To utilize creativity for innovations, there is a need to locate and deploy it within the process, i.e., ideation, development and commercialization, with innovation being an output thereof. The creative process goes further than the simple production of visual outputs, as design is inserted into many areas of management decision-making. Design (process) is an internal management process that integrates market research, marketing strategy, branding, engineering, new product development, production planning, distribution and corporate communication policies [131] (p. 18). Furthermore, the design process can facilitate he innovation potential of a conceptual solution with a set of elements, so-called innovation vectors, and improve the decision-making process for a given design problem [132] (p. 59). In addition, it might reveal new avenues for companies to consider and integrate preferences of customers and experts for product innovation through the prioritization and evaluation of product design factors [133] (p. 13). The recent literature confirms that innovation is an outcome of scientific activity and creativity, and that the combination thereof is a key to innovation.

Indeed, this correlates with already existing and confirmed interdependencies stemming from neo-classical discourses in terms of place-based approaches and thus institutional aspects on CCIs: (a) agent cognition and learning; (b) social networks, and (c) market-based enterprise, organizations and coordinating institutions [134]. Since CCIs address market contexts that are much closer to the extreme of networks' effects, CCIs can be referred to as economics of networks (pp. 170–171). Whereas the processual dimension refers to CCIs and, in particular, creativity, implying a process of discovery, S3 design and implementation is subject to entrepreneurial discovery process (EDP) [135]. CCIs are social constructs entailing interactions and creating social and shared value, while S3 legitimacy is also based on successful, open and experimental interactions of quadruple helix actors, as discussed above. Indeed, the importance of social aspects, e.g., social networks on the regional and local scale of CCIs and their relationship with the location, has been ignored for a long time [136] (p. 718), [137] (pp. 162–163). Yet, helix models are crucial, since they understand innovations as complex processes embedded in the nexus of institutional agents and cultural aspects [138] (p. 5), thus making proximity an important precondition for both knowledge generation, exchange and transfer [139] (pp. 1–2). Since creativity is developing across the whole economy, the inclusion of the quadruple helix model is crucial [140].

In this sense, by acknowledging CCIs as knowledge mediators, brokers and coordinators [141], as well as essential enablers in innovation development, they can become principal agents within quadruple helix partnerships for value creation, thus generating innovative ideas and providing novel research trajectories [142] (p. 137), also in the nexus of micro-level value creation mechanisms [143]. Indeed, the engagement of CCIs within quadruple helixes could also improve their functioning. Currently, they are not efficient enough, based on the complexity of interactions, conflicting logics, prioritization needs and compromises of value [144] (p. 15). Collaborative governance, exemplified through the involvement of multiple actors or collective actions, supports S3 implementation, in particular by means of collective knowledge generation and learning, along with endogenous competences [145]. Collaborative networks are an indispensable tool to improve idea generation and accelerate positive results of the creative process based on the expertise diversity of the involved social actors [146]. CCIs are also driven by demand, diversity, locality, education and skills, networks, the public sector, and business capacity, which, in turn, build up classical helices [147] (p. 344). CCIs aims at creating new or refining old.

Innovation-oriented S3 aims at policy implementation and improving regional innovation systems, with innovation being at the core. Bearing the aforementioned in mind, it is pivotal to strengthen both the conceptual as well as the practical alignment of CCIs, S3 and sustainability within place-based discourse, as already demonstrated in preceding scholarly entries.

## 3. Materials and Methods

The present work presents itself as an interdisciplinary research project that seeks transdisciplinary solutions by linking different competences. Its interdisciplinary nature was already clearly delineated in the introductory and theoretical parts, by linking up CCIs for S3 within the sustainability nexus and regional setting. Creativity and sustainability can be approached from different transdisciplinary and cultural perspectives [61]. Methodologically, the research builds upon actor-network theory [148] and a social system-based perspective [147], including the recording and evaluation of cooperation patterns in the given peripheral region of the SBSR. The actor-network theory (ANT) provides a research framework that enables one to explore dynamic and socially constructed phenomena and their interactions [148] (p. 1026). Indeed, for the present research, due to its complexity and the intertwining of different social constructs, it appears to be feasible to choose this research framework. In addition, this approach is suitable for project management purposes [149]. Given the presence of the applied research project "CTCC", which provides the fundamentals for the empirical inquiry, ANT fits well here in order to trace the intertwining associations between human participants, objects and processes at all levels of the project [150,151]. Indeed, social interactions are usually studied on the basis of empirical case studies [109] (p. 43). Moreover, S3-related exploration also advocates for institutional learning in place-based pilot projects [144] (p. 16).

Since this work aims at contributing to future projections and explores regional potential, it is evident that future research should remain problem-oriented and participatory. The latest studies emphasize that nowadays, scientists should choose super-disciplinarity as a research approach and collect new findings from different research fields in order to look at the topic holistically. Indeed, in the nexus of S3 and EDP, self-discovery should be open to techniques known from cultural anthropology [121] (p. 1808), e.g., participatory observation [152], as a way of developing insights through contexts and first-hand experiencing [153] (p. 238). Here, social anthropological processes should be strengthened in the research, in particular in qualitative research, within which connections, interdependencies, components and dynamics in the given ecosystem are investigated [154] (p. 17). Indeed, S3-oriented research needs to go beyond quantitative studies and provide room for new paths of deploying qualitative research methods in different regional contexts [127] (p. 196), [155] (p. 1642), [156] (p. 1). Overall, this work adopts an exploratory qualitative approach [157,158]. Taking the research gap into consideration, following Creswell (2014), the authors highlight that if a concept or phenomenon needs to be explored and comprehended, since only scant research in this area has been performed, then a qualitative approach appears to be feasible. Additionally, qualitative research is especially useful when the researcher does not know the important variables to examine [159] (p. 50). Therefore, the impetus of inductive reasoning and conceptualization qualifies the use of this overall methodological approach [160].

An action research approach was employed throughout the entire research trajectory [161–164], as the research lasted over a longer period (2020–2021) and the research results were recalled and reconciled in several progress phases. This approach fits the present research effort, since it is able to provide a way to act in a holistic and complex way. It supports the dualistic and dialectic view employed here and discussed above, as well as opening up opportunities to bridge both science and practice. Furthermore, it enables one to intertwine different research methodology categories [165] (p. 151). This is of particular importance, since the researchers of this paper were directly involved in the ongoing project as innovation brokers—creative brokers—and undertook an observation and assessment

of the innovation phenomena concerned. Preceding learning cycles were integrated into the upcoming research activities, which, in turn, also constitute an important research component [166] (p. 281).

The present research deploys a case study. The multi-case study is based on the Interreg V A project "CTCC—Creative Traditional Companies Cooperation" (2017–2021) (rf. Table 1). The CTCC project investigates cooperation models between creative and traditional companies by developing cross-sectoral innovation in the product, service, organizational and marketing areas (here, the demand of creativity by traditional small and medium-sized enterprises—SMEs) in different industry sectors within the S3 policy in the SBSR. For the purpose of the present research, individual cross-sectoral collaboration patterns resulted in the development of individual innovation prototypes in the concerned sectors, in which quadruple helix actors, such as researchers, CCIs, traditional businesses and policy makers, were involved. In total, the innovation journey yielded 33 prototypes for 32 traditional SMEs achieved in the time frame starting in April 2019 and finishing in September 2021. Out of 33 prototypes, seven concern innovation efforts in the marine and coastal tourism sector. As a result, seven topical prototypes were subject to this research, and thus build up the empirical body. Therefore, the sampling of this research is purposive [167]. The availability of the seven individual cases (innovation prototypes) enable within-case as well as cross-case analysis, following case study content analysis [168,169]. The cases refer to a new phenomenon, and therefore they rely on abductive reasoning embedded in empirical data and bridge rich qualitative evidence to mainstream deductive research [170] (p. 25). Thus, a methodological conceptualization builds up the first step of the overall research design and enables us to analyze and evaluate CCIs potential for sustainability and resilience of tourism sector in the S3 policy nexus. This is consistent with future research efforts that should be made in terms of S3, namely exploring micro-level and place-based effects in quadruple helix partnerships [171] (p. 1070). In addition, this work aims at delivering not only theoretical but also practical insights and recommendations as a result of the applied research project. Therefore, pure conceptual arguments are underpinned with illustrations. In this way, the research aims at showcasing the real world and not just the literature [172] (p. 23). Overall, the research focuses on current phenomena and addresses research questions of "why" and "how" [173] (p. 2) (rf. Introduction), [174] (pp. 4–6), thus underpinning the rationale of the overall case study methodology utilization, which is highly recommended to perceive behaviors in intersectional relationships in network studies [175].

**Table 1.** Research setting and case allocation.

| Research Design Item | Research Response |
| --- | --- |
| Research scope | Interreg V A project CTCC—Creative Traditional Companies Cooperation |
| Geographical coverage | South Baltic Sea Region—Danish, German, Lithuanian, Polish and Swedish coastal regions |
| Research scale | Seven innovation prototypes for marine and coastal tourism |
| Research approach | Inductive |
| Research methods | Shadowing, co-creation, expert interviews, participatory observation |
| Research data | Qualitative |
| Research techniques | Innovation prototype analysis, cross-case (prototype) comparison, template analysis, self-discovery |
| Research validation | Quadruple helix experts, customers, external experts |

Source: compiled by the authors, own illustration.

The case study methodology of this work followed the empirical triangulation approach [176–178], and aligned such empirical qualitative methods as (a) seven individual innovation prototype cases of the project SMEs, followed by innovation development utilizing individual shadowing, supported by data provided by innovating SMEs and other

quadruple helix actors concerned with the innovation under development; (b) participation in innovation sprints aimed at recalling and revamping the innovation development progress of each SME through co-creation; (c) 32 expert interviews conducted during and after innovation development projects in 2020–2021 and linked to challenges and sustainability efforts of SMEs; and (d) participatory observations of workshops and focus group meetings.

Due to the outbreak of the COVID-19 pandemic, some of the interviews were undertaken in online formats [179]. The triangulation or double-source methods reduced biased information, in particular through the validity and heterogeneity of the multiple data, and therefore necessitated a full understanding of a complex contexts and intertwined phenomena [180–182]. All collected data and participatory information were subject to content analysis [183–187], by deploying thematic coding based on the anticipated research aim following interdependencies of CCIs, sustainable and resilient tourism in the S3 nexus. The research journey encapsulated the following process steps, such as participating in the project, developing templates for data gathering, gathering the data (field research, cases analysis, expert interviews, secondary data), decoding, analyzing the contents and synthesizing, amalgaming and iterating the empirical results with the literature. Consequently, conceptual insights for the scholarly community and practical considerations were elaborated on, followed by positioning the present research within future and research avenues.

## 4. CCIs for Sustainable and Resilient Tourism in the SBSR within S3 Policy Discourse

Following research analysis and evaluation, this paper presents research results and showcases CCIs' potential for designing and enabling sustainable and smart regional development based on the purpose sector–marine and coastal tourism. Since this sector represents one of the key economy drivers of the South Baltic Sea Region (SBSR) on the one hand, but bears a series of risks due to the lockdown situation, the inability to step out from the depression and the missing capacity to respond to new technological trends and the increasing impact of a twin (environmental and digital) transition in the frame of the European Green Deal (EGD) on the other, searching for and finding potential new innovative pathways represents a substantial step forward. A second step can be linked to the research effort to explore the potential of peripheral lagging regions within the EU innovation policy framework, in particular by aligning it with the S3 policy for 2021–2027 in terms of exploring innovation potential and shared value creation involving quadruple helix actors and shedding light on micro-level collaboration patterns and their results. The results are presented, first, by yielding CCIs' interventions for sustainability and resilience in place-based innovation projects, followed by, second, a reflection upon those results and their impact on S3 policy design and implementation in terms of micro-level constellations in processual and institutional arrangements, and, third, by adopting innovation prototyping tools co-created with the CCIs to streamline S3 policy.

### 4.1. Exploring CCIs' Potential for Sustainability and Resilience via Regional Innovation Cases

As already mentioned above, the research sampling is purposive, and seven innovation prototyping projects are referred to as cases within this research frontier (Table 2). This is traced back to the project's nature, which is characterized by a variation of priority areas (S3 priorities) chosen, e.g., ranging from sustainable food processing to offshore wind energy, marine and coastal tourism, ship building and marine transport. Since the tourism sector represents a key building block in the region under scrutiny, for a better positioned purpose, only cases from marine and coastal tourism were selected. However, due to the sectoral linkages of tourism with other regional industries, such as nutrition, mobility and transport, other cases were interlinked for comparison purposes.

**Table 2.** Tracing sustainability and resilience in the SBSR tourism sector through CCIs' intervention.

| Case ID | Case Maxim | Location (NUTS-2) | CCIs Potential for Sustainability | CCIs Potential for Resilience |
|---------|-----------|-------------------|-----------------------------------|-------------------------------|
| DE1 | Plastic free City of Rostock | Mecklenburg-Western Pomerania | Environmental Institutional | Preparatory Adaptive |
| LT1 | Sustainable rural forest tourism | Central and Western Lithuania | Environmental Economic Social | Preparatory Adaptive |
| LT2 | Solar-energy driven water ride | Central and Western Lithuania | Environmental Economic Social | Preparatory Adaptive |
| LT3 | Improved marine related festivities | Central and Western Lithuania | Social Institutional | Recoverable |
| PL1 | Gamified group trip offer | Pomorskie | Environmental Social | Absorptive Adaptive |
| PL2 | Online Platform as way of visiting historical sites | Pomorskie | Economic Social | Absorptive Recoverable |
| PL3 | Attractive children water sport | Warminsiko-Mazurskie | Economic Institutional | Preparatory Absorptive |

Source: compiled by the authors, own illustration.

According to the content analysis of the innovation prototypes (cases), CCIs' engagement within the innovation development process with each single SME reveals both sustainability and resilience attributes. Within this research, the authors differentiate between the potential of CCIs for sustainability and resilience, although there exist myriads of research streams, mainly driven by discourses about resilience as a component of sustainability, sustainability as a component of resilience and sustainability and resilience as separate paradigms [188] (p. 1275). Recalling the contextual intertwining of CCIs and S3 in the sustainability nexus expressed through process and institutional dimensions, this research builds upon the precept of separating sustainability and resilience spheres. This contributes to better embeddedness of CCIs in both sustainability and resilience-related discussions. In addition, it is believed here that, whereas sustainability addresses social and institutional aspects, also raised within S3 policy design and implementation, resilience refers to capabilities built up in the time lapse (evolutionary). For this, CCIs can reveal both institutional and process-driven benefits in the regional setting.

Building upon a synoptic view of the cases, it can be contended that CCIs' intervention leads to improved sustainability performance in SMEs and at the regional level, in particular leading to enhanced environmental consciousness, social responsibility, improved economic efficiency and institutional thickness (social coherence). CCIs' contribution in innovation projects is evident as initiating the changing of minds, advocating for the growth of integrity, reviving connections with and the recognition of customers and users, boosting circularity, and promoting regional identity and adaptive capacity. In doing this, CCIs are clear contributors and promoters of resilience capacity building. Following innovation development projects with CCIs, participating SMEs are better at absorbing the potential impacts, recently, as a result of the COVID-19 pandemic, recovering through new pathways detected in the frame of co-creation with CCIs and being better equipped for any potential future environmental, economic, social and institutional shocks and disturbances. This is very true for the present research. Despite the COVID-19 outbreak, which disrupted individual innovation development by SMEs in collaboration with CCIs, all initiated innovation projects were accomplished, although with a delay, thanks to CCIs' intervention and the continuation of partnering and support for traditional SMEs. Moreover, CCIs also supported traditional SMEs from the region in the face of the newly released European

Green Deal Strategy and anticipated the engagement of all economy and social agents in the environmental and digital transition.

In terms of environmental sustainability efforts, the DE1 case SME (local food service provider for locals and tourists) engaged in the prototyping process with a focus on a new initiative, namely, to move the city of Rostock towards the status of a plastic-free city. For this, the network-based entrepreneurial initiative aimed at the reduction in the consumption of disposable plastics. To make this concept real, the DE1 case SME needed to overcome the main challenge of finding, engaging with and securing long-term sustainable behavior and coordination, first by reducing disposable plastics in a self-responsible and self-organized way within its own organization. In addition, the succeeding question was how sustainable behavior could also be achieved among customers by actively engaging them into engaging with the alternative and sustainable waste disposal mechanisms. Together with CCI representatives, mainly graphic and web designers, the DE1 case SME was able to develop a toolbox consisting of three instruments, i.e., an individual tool (website with manual and calculator for disposal plastics), an internal expert tool (workshop) and a networking tool (enabling interlinkage with other entrepreneurs). At the end, all tools were transformed also into a digital format, in the form of an explanatory video illustrating the essence of each of three instruments. By doing this, the DE1 case SME was able to unveil its sustainable potential through intervention with the CCIs, as they helped the public to understand and perceive not only internal, but also external (environment, tourists, city citizens) needs and expectations. Moreover, CCIs contributed to making the location (city) as well as the entire region environmentally conscious by strengthening target group (tourists and citizens) participation, delivering digital implementation formats (such as videos) that enable a broader reach out. With these sustainability efforts, CCIs drove co-creation, facilitated planning and adaptive capacity building for a changing market and societal behavior, such as EU plastics bans in force or in the future, generated societal movement towards unpacked products, and provided discounts for disposable plastics, etc.

A similar environmental sustainability footprint was achieved in the LT1 case. Within this case, the SME faced higher competition and challenges to attract tourists to the region, since it is not directly located on the Baltic Sea, but in the periphery of the coastal region of Lithuania. For this reason, providing an attractive touristic destination beyond the coast bears a clear challenge. For this reason, in the collaboration with CCIs, the main aim was making the touristic destination in the forest area attractive through ecological and social consciousness, i.e., making the stay in the forest use as few resources as possible, i.e., staying in a wooden house by the water without any Wi-Fi, electricity and water supply, using a zero waste practice and solar energy, and thus becoming part of nature on the one hand as well as improving human health conditions on the other. The co-creation resulted in the business model "forest adventure", claiming higher value proposition and positioning it in a higher price segment, due to the unique touristic location. With the creation of the new business model, opportunities for diversification via franchising into other regions was also seen as a feasible business option and additional business growth alternative, thus fostering employability and job creation opportunities. As a result, the prototyping process delivered a business model that can be used for green slow sustainable tourism, providing revenue and at the same time having a social function, including inclusion and gamification, because potential tourists receive the address of the location just before arrival. Moreover, local artists and small craft service providers are involved on the road to the location, either directly or indirectly. Creative brokerage, service design methodology and the incorporation of smart technological solutions into the business model appearance (design) (online platform, video) made it possible to address both environmental, economic and social dimensions of sustainability and to prepare for the zero emissions economy that must become a reality by 2050.

While the LT1 case SME focused on innovating rural tourism, the LT2 case SME aimed at strengthening environmental performance of water tourism in the region/city of Klaipeda. Belonging to the EU coastal region, Klaipeda is also encouraged to comply

with all environmental regulations regarding transport and mobility, as well as to engage in initiatives fostering marine resource conservation and contributing to blue growth. The main challenge for this SME was missing graphic design skills and technological capacity in transferring the ideas into a feasible form. While having a long-term goal to become a mind changer in the cargo shipping industry in the region and beyond it, the LT2 case SME co-developed with CCIs a solar-powered catamaran designed to be used in urban waters for sightseeing, transportation and leisure activity. The prototype consists of two parts—visualization (form) and 3D printing of the prototype. As a result, the innovation development project yielded the benefits of simultaneously providing functionality and interacting, facilitating communication and engaging with customers, users as well as potential investors more actively through the activated human senses (see, touch), improving the efficiency of product design through the 3D printing outcome using ecological materials (corn plastic) and other construction parts made of wood and metal, reducing material wastage, product development time and production costs. One of the main disadvantages is having a not real-size prototype, as it was reduced 22 times in scale. Nevertheless, the innovation prototype represents the functionality and appearance of the anticipated final product, in addition saving energy and material sources for its printing. Overall, the joint cross-sectoral collaboration between the LT2 case SME and participating CCIs (graphic and service designers) contributes to environmental, economic and social aspects of sustainability, principally through cost and time reduction, material savings, improved product process efficiency, and increased quality and customer satisfaction. Building up preparatory resilience capacity in terms of upcoming environmental regulations and in the face of increasing environmental compliance for the tourism industry can be expressed through CCIs' intervention in providing and sharing knowledge relevant for environmental consciousness and customers' environmental thinking, translating complex product data into simplified forms, unlocking decision-making and learning processes, providing opportunities for early testing and analysis, enabling the testing of futuristic contents, evaluating safety-critical risk management tasks and improving social conditions (reducing $CO_2$ emissions and making the region more attractive). By virtue of this innovation prototype, new environment-driven business diversification opportunities can be linked with this disruptive innovation coming to the region, such as renewable energy providers and technological component suppliers, the attraction and settlement of environmental experts in the region, as well as follow-ups by other tourism service providers.

Compared with the preceding innovation prototyping case, a softer tourism-oriented innovation built upon the idea of service innovation was the focus of the LT3 case SME aiming at delivering a service solution for the city of Klaipeda, namely providing a platform for interaction between local craftsmen and artists together with citizens and city guests for the purpose of idea/product generation and creation during the biggest regional and national event—the sea festival—taking place each year. The LT3 case SME (public institution), which is responsible for the implementation of the entire sea festival, engaged in a collaboration with CCIs supporting the initiative of co-creation and co-discovery. For this, analysis of the local/regional position in the tourism market, market segmentation and market needs helped to explore, evaluate and communicate the service prototype "art meadow" as a collaborative, explorative and iterative activity for families and city guests aiming at the active expression of their ideas or searching for new opportunities for talent identification. CCIs contributed with the visualization of the prototype and content generation articulated through workshop organization and implementation in the frame of the sea festival. The main contribution of this innovation case to the regional tourism sustainability is expressed through social and institutional dimensions, in particular by enabling and strengthening the dialog between CCIs, citizens and tourism, thus generating an active participatory tool for learning and expression, which, in turn, leads to the underpinned institutional engagement of local CCIs in place branding and place destination initiatives, simultaneously also strengthening local and regional identity and wellbeing. With this intention, CCIs' intervention in the local/regional touristic initiative can provide local and

regional actors with the resilient capability to recover from the disturbance, such as low local CCI involvement in touristic activities despite the strong creative orientation and expression of the sea festival as the main regional and national cultural attraction.

Linked to marine and coastal tourism and the better utilization of marine resources is the innovation case PL1 and the Polish SME, which aimed at improving the recognition and ecological footprint of group travel services dedicated mainly to businesses (employees, contractors and business partner groups), thus also closing the gap between touristic offers that fit into a strong ecological and sustainable mindset. With this offer in mind, the PL1 case SME aims at adapting to the changing behavior of professionals, who usually seek not only economic but also social value creation going beyond attractive salary incentives. In doing this, within the process of co-creation with the CCIs, an interactive platform in the form of a quiz aimed at adapting the group trip offer to the users' needs was generated as an innovation prototype, leading to new service offered in the tourist service sector. In order to better explore the needs and expectations of a group of persons aiming at a joint trip, the interactive platform adjusts a potential cruise trip to the explored needs and expectations of a group of persons, following the quiz. As a result, the service innovation facilitates the personalization of a trip for a group in an accessible and intuitive way, by using questions and algorithms, resulting in an available tailor-made touristic offer package. This, in turn, provides a unique selling point for potential customers and users, since the service is personalized and attached to the group's needs. Personalization tracking and expression is supported through functional and graphic solutions offered by the CCIs, in particular showing possibilities and types of cruise formats, such as meetings, training, and other incentives in visualized formats (a form), thus also increasing customer bonding through the showcasing of opportunities and curiosities attached to the projected group trip. Overall, this innovation prototype claims CCIs' contribution primarily in enabling social inclusion and strengthening bonding among employees of one group aiming at one trip on the one hand, as well as better customer recognition through the personalization of activities on the other. In addition, environmental traits are visible through the focus on ecologically conscious behavior. The observed innovation journey of the PL1 case SME pinpoints absorptive and adaptive resilience capacity building, as changing customers'/users' preferences in the tourism and overall economic performance paradigm enable the SME to absorb and integrate their needs and expectations ex ante any potential disruptions to the market (acting in advance, rather than reacting too late) as well as to adapt to changing tourism business paradigms as a result of both new environmentally and human (behavior)-driven interactions.

Whereas developing a group trip innovative service might help in tourism company business diversification and building up competitive advantages through direct customer targeting (personalization) and entering niche markets (business trip focus), as illustrated in the PL1 innovation case, the internationalization of local products of craftsmen, artists and designers who build upon tradition continuation represents a clear challenge in the highly touristic region of Gdansk, and this is the challenge faced by the PL2 case SME (tourist shop). Moreover, due to increasing overall pressure on tourism, bringing crucial disturbances and a recession as a result of the COVID-19 pandemic and the post-pandemic depression, the PL2 case SME had to cope with pivotal challenges, such as price changes, increasing awareness raising needs and thus costs, and changes to online selling opportunities, that, unfortunately, resulted in the physical closure of the giftshop. By engaging CCIs who are known for being digitally savvy, an online platform was sought as an alternative for business operation and strategic positioning. The main aim was building up a giftshop brand promoting specific values, namely community feeling, such as the handicraft work process and its importance for the region and the origin of the products offered. As a result of the co-creation and innovation prototyping process with CCIs, the PL2 case SME attained a twin business model: while the focus should be kept on developing an online shop, its uniqueness should also build upon a design guide promoting local craftsmen, artists and places worth seeing in Gdansk and its surroundings, thus making it an alternative to other

giftshops as a platform providing not only products (gifts), but also access to other tourism services, such as alternative city tours, alternative places to visit and so on. Moreover, additional selling points of this service/business model can be linked to the promotion of local brands highlighting local handicraft as an alternative to mass production. In sum, this innovation prototype serves as an example of CCI intervention supporting both economic and social sustainability, in particular by enabling the PL2 case SME to find alternative ways out of the lockdown and economic tourism shocks, as well as by promoting regional identity and common belonging through the utilization and facilitation of local natural and cultural resources (local handicraft). As a means for the breaking through the COVID-19 pandemic shock, the PL2 case SME is able to strengthen its resilience capability, in particular by absorbing the impacts of disturbances (COVID-19) and recovering through alternative business models.

Local recognition challenges also hamper the PL3 case SME in business expansion and market penetration. This SME, producing kayaks for children, aimed at improving supply chain linkage, specifically through engagement in new supply chains, e.g., by entering new business collaboration with a bigger company able to produce kayaks in a desired format for the PL3 case SME. The main CCI contribution was made through the visualization of the needs and desires as expressed by the SME, following market research and customer journey. Market niches were recognized during the innovation prototyping, as no other companies were found that produce kayaks for children, and if so, their offer on the market was not diversified (only one option to choose, both regarding color and form). As a result, CCIs helped to provide the competitive edge of the product and to transfer it to a more competitive status, mainly driven by customer/user engagement and the recognition of their needs (children choose personalized and colorful products reflecting their own personality). A tangible innovation result was a product prototype visualized with the creative means in three different forms and colors transferring the meaning of character and behavior, such as an octopus, whale and dragon. Following the gamut of potential forms of children kayaks, the evaluation of the participation of customers was also undertaken, thus supporting the positive acceptance and meeting expectations. Overall, the implementation of this innovation prototype can increase touristic and leisure attractiveness through the inclusion of the young generation, while at the same time strengthening the water-based touristic attractiveness of the region (water sport and leisure activities) and the regional identity in general, thus highlighting the social aspects of the sustainability concept. From the economic point of view, CCIs' intervention supports the diversification opportunities of the project. Institutionally, engagement in new collaboration patterns and thus enhancing participation and recognition in the supply chains contributes to strengthening institutional sustainability. The social, economic and institutional aspects and their interplay helps the PL3 case SME to prepare in advance, by increasing competitive edge as well as adapting to changing/arising new customer product selection behavior.

From the synoptic point of view, all illustrated SME cases show improved sustainability and resilience potential through CCIs' interventions. Therefore, this research clearly supports CCIs' potential for building up sustainability capacity and resilience capabilities on the local and regional level based on the utilization of local (marine and coastal) resources for touristic and recreational purposes. In sum, co-creation among traditional (tourism sector) SMEs and CCIs contributes to both providing capacity and capabilities for the SMEs to find new means for renewal as well as utilizing new business opportunities, transferring regional businesses towards an environmentally conscious, resource-efficient (economic), socially inclusive and institutionally stable status, showing clear tangible contributions to the local and regional economy.

### 4.2. Amalgamating and Consolidating the Potential of CCIs for S3 Policy Development Pathways

Bearing in mind the analysis and consolidated evaluation of the available innovation cases of SMEs resulting from innovation development prototypes, it is evident that CCIs' intervention can be traceable not only for improving the regional innovation capacities

and capabilities of local and regional actors, but also for strengthening the sustainability performance and resilience capabilities of involved economic agents. Indeed, since sustainability and resilience represent anticipated goals to be reached with regional innovation policy (re)direction, in particular S3 for the upcoming EU cohesion policy period 2021–2027, based on the results of the present research work, a clear interlinkage can be revealed with CCIs' role and contribution in the renewal and revival of the S3 policy. As noted by several scholarly entries, S3 policy design and implementation have so far missed the exploration of potential and effects on the micro-level (partnership, firm-based). Instead, the focus has largely been on contextual analysis (linking up with regional performance, diversification, policy advocacy and battling over sectoral/thematic prioritization) and the quantification of S3's effects for regional and national economies. In contrast, the present research contribution reveals the untapped potential of CCIs and their intervention, which within the pool of the seven innovation cases analyzed is strongly linked with the social (five out of seven cases), environmental (four out of seven cases), economic (four out of seven cases) and institutional (three out of seven cases) sustainability interplay. In terms of resilience dimensions ranging from preparatory/planning to absorptive, recoverable and adaptive capability, following existing conceptual taxonomy [188], CCIs' intervention in seven innovation prototypes contributed to improving and enhancing the preparatory and planning capability of participating SMEs (four out of seven cases), followed by their adaptive (four out of seven cases), absorptive (three out of seven cases) and recoverable (two out of seven cases) capabilities.

The interpretation of these results can be manifold. However, it is clearly visible from the used innovation cases that CCIs are future-oriented, aiming at providing SMEs with tools and capacities to build up capabilities that will drive future business and thus strengthen strategic position, e.g., through specific contributions to adaptive capability and the ability to absorb future needs and expectations of the market (customers, users), as well as changing environmental conditions (through new environmental regulations and environmental/social/legal compliance). Moreover, although CCIs helped participating SMEs from the tourism sector to break through the recession and depression as caused by the COVID-19 pandemic, their potential for recovery remains limited. This can be traced back to several reasons, starting with the fact that the majority of innovation pilots were started before the outbreak of the pandemic, thus leading to the assumption that the majority of participating SMEs already had a clear idea about the innovation journey outcome. Alternatively, the available number of the SMEs cases and their unique characteristics (bound to the local/regional setting) might not be enough to ground this reasoning. In addition, since the tourism sector was heavily hit worldwide and also in the region (it is the most important competitive advantage of the SBSR), it can be assumed that the vast number of the sectoral SMEs participating in the project were coping with overcoming usual (daily and running) business obstacles and challenges, while focusing on positive/future-oriented business opportunities in the frame of innovation projection, which by its nature is future-driven and carries in it futuristic connotations.

Building upon the sustainability and resilience lens and the reflection of CCIs' contribution to the micro-level (firm-level) as well as bearing in mind the demonstrated micro-level impacts on the local/regional setting (environment, expressed through ecological, economic, social and institutional arrangements and their interdependencies), the assessed sustainability and resilience impacts were synthesized and consolidated within the S3 policy nexus, i.e., by positioning the effects within the place-based setting—NUTS-2 regions in the EU (using official NUTS nomenclature coding applicable for respective region identification), in which innovation prototypes were explored. For this purpose, individual case-specific analyses and evaluation attributes were reflected through individual weighting of sustainability and resilience contributions by CCIs on the regional level. Whereas the achievement of all four sustainability (social, environmental, economic and institutional) and, respectively, resilience traits (preparatory/planning, absorptive, recoverable and adaptive) can be expressed by four plus "++++", each missing sustainability dimension out

of the maximum four available leads to one plus less, thus arriving at three, two, one or, consequently, zero sustainability and/or resilience anchoring. This weighting gamut was applied based on the available innovation cases (one to seven).

Following this classification, other proximity types were used as parameters to trace and evaluate CCIs' contribution to S3 policy design and implementation, following Boschma's proximity taxonomy [189,190], such as institutional, social, cognitive and organizational proximity. Based on existing knowledge ties and their effects within and on knowledge networks, a different proximity might result in possible learning, decoupling, institution-alization, integration and agglomeration process effects. By linking up with common conceptual traits established between CCIs and S3 policy and rooted mainly in processual (invention (creativity) and development driven) and institutional (partnerships and networks) arrangements, CCIs' potential can also be seized within S3 process (entrepreneurial discovery process (EDP)) and institutional (quadruple helix networks and collaboration patterns) arrays.

As a result of the analysis undertaken, CCIs' embeddedness in both dimensions can be evaluated by deploying the following scorecard: in terms of processual-level arrangements, CCIs' contribution can be registered through creativity contribution towards the EDP, in particular innovation process efficiency, effectiveness and correlation, expressed through the holistic self-discovery process leading to tangible results (output), through stakeholder engagement (1), ideation/intervention (2), development/design (3), realization/validation (4) and commercialization/capitalization (5). Within the given innovation cases, EDP or self-discovery was achieved through the development of tangible innovation solutions (prototypes). The intensity and/or contribution of CCIs towards the realization of this innovation process can be measured by applying a range recalling quadruple helix actors' interactions and engagement within this five-step process, thus meaning that five plus "+++++" stand for a holistic step-by-step self-discovery process passing all of the above-mentioned phases and engaging all affected environment (ecosystem) actors, such as SMEs, researchers, policy makers and society representatives, who through their successful collaboration patterns can lead to the desired overall sustainability, delivering environmental, economic, social and institutional advantages in that particular ecosystem. Respectively, each plus less implies a less intensive and successful process. In terms of CCIs' potential for institutional arrangements, a similar scale is suggested: whereas the achievement of institutional proximity driven by CCIs' intervention articulated in terms of cognition and learning (1), decoupling (renewal of opportunistic behavior and unblocking/loosening entry barriers) (2), institutionalization (3), integration (4) and agglomeration (5), in total resulting in a potential five scores, can be expressed as five plus "+++++", less efficient knowledge network effects missing one of these five potential social constructs lead to the succeeding deduction of each individual plus. Table 3 presents the yielded evaluation according to the presented classification for each case as well as the respective NUTS-2 region.

Overall, the presented overview in Table 3, by avoiding any repetition of the already analyzed and assessed innovation cases in the previous section, provides a consolidated evaluation of CCIs' potential for S3 policy design and implementation. In detail, innovation case implementation in the German region (DE80—Mecklenburg-Western Pomerania) confirms the achievement of environmental and institutional sustainability (++) as well as preparatory and adaptive capability (++), strengthening local and regional resilience. In terms of CCIs' intervention within processual and institutional arrays, although the innovation prototyping process involved important actors from the quadruple helix partnership, such as SMEs and society (target groups), the innovation journey rather missed policy intervention, which is crucial in terms of environmental (legal) compliance and has high-level impacts on environmental regulative side, thus making it possible to either positively or negatively affect the outcome of the innovation prototype. Considering the fact that the empirical foundation for the German region is grounded in this one innovation case, it should be noted that the overall evaluation of this region might fall behind the

performance of the studied Lithuanian (three cases) and Polish regions (two cases from the same NUTS-3 region).

**Table 3.** CCIs for S3 policy design and implementation in process and institutional arrangements.

| NUTS-2 Region | CCIS Intervention in S3 Yes/No | | CCIs for Sustainability | CCIs for Resilience | CCIs Potential for S3 Implementation in Micro-Level Process and Institutional Arrangements | |
|---|---|---|---|---|---|---|
| | 2014–2020 | 2021–2027 | | | Process Level | Institutional Level |
| **DE 80** | No | No | ++ | ++ | +++ | +++ |
| **LT02** | Yes | n/a | +++ | ++ | ++++ | ++++ |
| **LT1** | | | +++ | ++ | ++++ | ++++ |
| **LT2** | | | +++ | ++ | ++++ | +++++ |
| **LT3** | | | ++ | + | +++ | +++ |
| **PL63** | No | n/a | ++ | ++ | +++ | +++ |
| **PL1** | | | ++ | ++ | +++ | ++++ |
| **PL2** | | | ++ | ++ | +++ | +++ |
| **PL62** | No | n/a | ++ | ++ | +++ | +++ |

Source: Compiled by the authors, own illustration.

Similar observations apply in terms of the rather distant involvement of researchers and academics, who can broker environmentally and socially important information through facts and value qualification. Moving towards the CCIs' potential for the Lithuanian NUTS-2 region (LT02—Central and Western Lithuania), the overall regional scoring is better than that in the German region, specifically driven by the fact that the participating Lithuanian SMEs from the same region take a rather opportunistic proactive approach rather than a reactive (regulation-driven, ex post) one when pursuing innovation pathways. Simply said, this means that environmental opportunities are recognized prior to environmental legal frameworks being put in place in this region, mainly by acting on changing customer/user needs and expectations. In sum, this enables this region to equip itself with environmental, economic and social/institutional sustainable business competence (+++) and prepare for any potential future disturbances by means of preparatory and adaptive capabilities (++) through engagement in joint innovation development projects with CCIs.

Finally, placing two Polish NUTS-2 regions' performance (PL63—Pomorskie and PL62—Warminsko Mazurskie) for comparative purposes, CCIs' intervention in the first region is traceable in particular in the environmental and social sustainability (++), pointing out the relevance and promotion of regional identity and place belonging, while at the same strengthening absorptive (+) capability, which, in turn, enables better recognition and integration of local/regional needs into innovation development, and thus potentially into the innovation policy design. From the processual perspective, CCIs' engagement leads to an overall holistic process, touching upon, in particular, environmental, social and economic aspects (+++). Social, cognitive and institutional proximity revealed by CCIs integration makes it possible to achieve knowledge creation and learning, integration and agglomeration. Regarding the latter region, CCIs are able to strengthen economic and institutional sustainability capacity (++). Through the demonstrated product's diversification potential, the strategic resilience capability can be strengthened (++), leading to improved social, institutional and organizational proximity (+++), as well as strong processual integrity, with less participatory integration of policy makers and research community into the innovation process, which, in turn, results in the not fully untapped stakeholder engagement and intervention within the innovation process.

Despite the fact that this scorecard might appear to be purposive and subjective at first glance, its postulation follows a thorough theoretical consideration, as presented in

the conceptual chapters that are both theoretically and empirically grounded. Potential improvement and advancement of this weighting scorecard can be anticipated through its application to a higher number of innovation prototyping cases (replicability and scalability), as well as its deployment beyond the diversified specialization, as in the case of tourism sector in this research work.

*4.3. Tooling S3 Policy Development with Creativity-Driven Potential*

Following the research results that confirm CCIs' intervention within co-creation and innovation development projects in cooperation with traditional sector SMEs from the SBSR (here, tourism), this research proposes two tools derived from the CCIs co-creation methodology that was developed in the frame of the CTCC project—(a) the Creative Audit Tool (Figure 1), and (b) the Creative Broker Concept (Figure 2). Respectively, these tools, once adapted to conceptual and managerial considerations within the S3 policy nexus, can be applied to S3 policy design and implementation, in particular in micro-level arrangements, in both anticipated processual and institutional dimensions, i.e., in order to improve interactions and collaboration patterns along the EDP and self-discovery governance on the one hand, as well as among quadruple helix partnerships and their actors on the other. As a result, building upon SME innovation case analysis and assessment, the developed tools can be adapted to the S3 policy context and be applied either on the entrepreneurial firm level (for innovation development)—policy makers and stakeholders' engagement, learning, achieving desired outputs (innovation, collaboration, capitalization)—or on the macro-level for local and regional planning purposes. Both tools are discussed from the co-creation (processual) and institutional (proximity intensity) perspectives.

Cross-sectoral, interdisciplinary and cross-institutional cooperation might lead to higher innovation outputs, as it has been seen in the SME innovation cases in the tourism sector, where all possible different perspectives are merged together to deliver value to customers and users, thus meeting their expectations and avoiding any unpredicted effects or negative implications on environmental, technological, market and social levels. This is achieved within the innovation development project, the so-called "Creative Auditing" (Figure 1), which is completed by interdisciplinary teams, where traditional sector SMEs cooperate and engage in innovation development together with start-ups, SMEs and freelances from the mentioned CCIs sector.

The Creative Audit Tool (Figure 1) was applied during the innovation prototype development phase that took place for almost two years in the CTCC project. In summary, the project applied a holistic and multidisciplinary approach, where several key tools and methods from the innovation development, creativity and design management were intertwined. The Creative Audit Tool supports the development of specific mechanisms and tools that stand behind the innovation processes [191] (p. 17). Indeed, the current research shows only a very limited record when it comes to practical tools for SMEs, e.g., the design audit tool by Moultrie et al., 2007 [192]. Yet, also existing, these tools mostly concern new product developments, but do not consider recent industry and transformation trends, i.e., digital transformation and Industry 4.0/5.0, as they were developed before industrial changes occurred. In addition, scrutinized approaches remain too theoretical and provide only auditing strategies without any specification of practical implications [193], static tools without a process orientation that deliver the status quo [194] or are oriented towards organizational culture only, i.e., branding and brand audits [195,196].

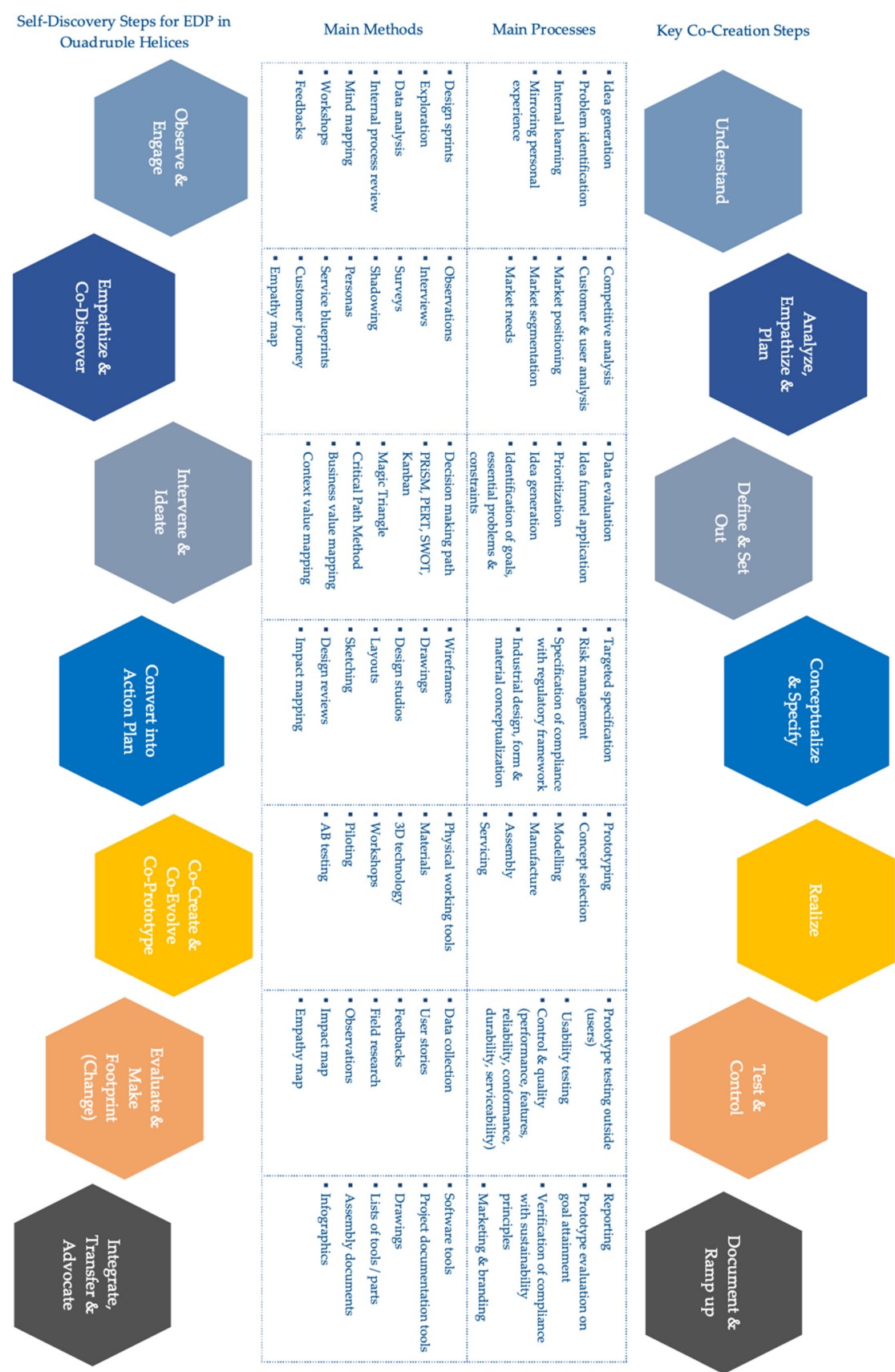

**Figure 1.** Creative Audit Tool for co-creation and CCIs' intervention in the S3 self-discovery process. Source: Compiled by the authors, own illustration.

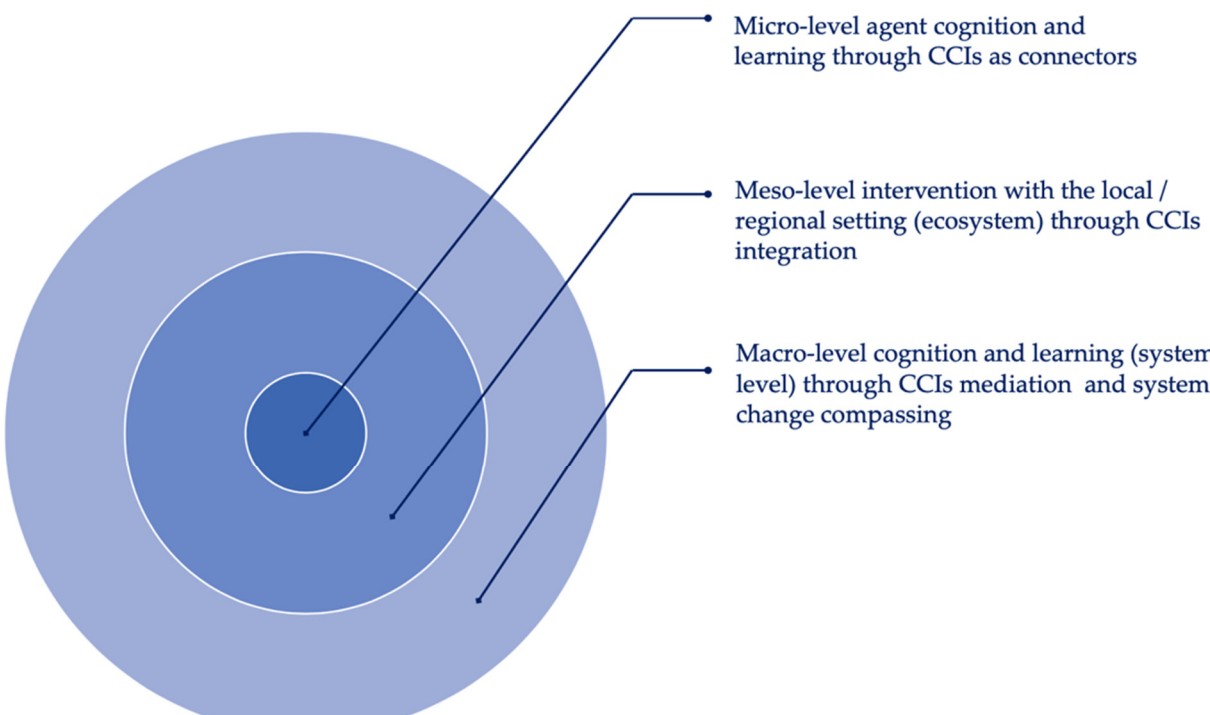

**Figure 2.** CCIs intervention (brokerage) for S3 policy institutional arrangements. Source: Compiled by the authors, own illustration.

By contrast, at the core of the CTCC, in the delivered Creative Audit Tool (Figure 1), there is an entrepreneur/SME that, by bypassing specific steps and applying given appropriate methods, is able to arrive at solid, tested and sustainable innovation output, be it a product, service, organizational improvement, e.g., change management or the application of new organizational procedures and methods, which optimize SME performance, or a business model aiming at improving the SME's positioning, differentiation and diversification efforts through targeted marketing and branding activities. Significantly, it differs from other similar tools, as it underpins innovation characteristics. Innovation is regarded as a key factor in transformation. By also integrating self-discovery traits from the S3 policy perspective, this toll can be simultaneously utilized for the integration and harmonization of the anticipated innovation process with S3 implementation. The tool becomes universal, and it can be used for any kind of innovation development, problem/challenge solving or company idea maturation.

Creative brokers act within the innovation process as "owners" of the Creative Audit Tool, who are responsible for guidance, counselling between creative and traditional sectors, policy makers and other quadruple helix partnership actors, also including innovation progress monitoring, control and evaluation. Since innovation is a process that requires high complexity, the tool (Figure 2) serves also as a facilitator and bridges diverse processes, actors and stakeholders meeting within innovation development. In particular, having an external source of innovation—a creative broker—a new perspective for improving institutional arrangements within the S3 policy can be introduced (Figure 2).

In order to be able to deliver innovation that enters the market and is used, industry SMEs need to reject their prejudices, learning other working cultures accompanied by different working languages, methods and environments. The same applies for other quadruple helix actors. Principally, an SME aiming at developing innovation should be ready to open up. As Wolfgang von Goethe expressed, "thinking is easy, acting is difficult, and to put one's thoughts into action is the most difficult thing in the world". Indeed, delivering what is desired by customers and users and what is innovative presupposes a high complexity and flexibility, diverse interactions and multi-faceted perceptions with the

readiness of a company for open-ended processes (as the final innovation output is bound to other processes, steps and interactions that influence it).

In order to act in this way, SMEs that usually lack their own resources or departments are able to supply creativity that is bound to external sources of innovation. In this light, a creative broker stands for a company or person engaging in the creative auditing and innovation development process. Rationally, creative brokers usually come from creative industries or possess expertise in innovation, design and creativity management. Creative brokers, as experts in knowledge, tools and methods transfer, facilitate cross-sectoral cooperation modelling, as they integrate creative, industry, policy, societal and environmental realms, advise SMEs in innovation development and manage interdisciplinary teams. Progress monitoring and quality assessment also belong to the task and competence portfolio of creative brokers.

By means of virtual or physical formats, e.g., speed-dating, job shadowing, and analysis of given SME structures, a series of exchanges and participation in the innovation prototyping process of SMEs (both physical and online), creative brokers act as innovation drivers, external observers and mediators within the innovation development process, where different perspectives, diverse expectations (company vs. customer/user), different resources and environments pull together. By doing this, creative brokers implement the so-called creative auditing by using the accompanying the tool, which enables us to understand SME structures and the behavior inside (organizational culture) and outside (market and ecosystem performance) the organization. Diverse methods and specific steps within the innovation process merged into the Creative Auditing Tool support Creative Broker in a progressive and systematic way, moving forward with the innovation development.

The engagement of creative brokers in innovation brings tangible value. They have a particular advantage in knowledge networks because they benefit from diversity at the levels of knowledge of different groups in the network. Creative brokering is not just the transfer of ideas from one source to another (as in the best practice knowledge dissemination, (rf. Gertler [197]); rather, it is the recombination of previous experience from various domains into new hybrid forms. Whatever the size of the agency or the project team, there are several key sites in the array of creativity that are at the root of this innovation [198] (pp. 685–686).

## 5. Discussion and Concluding Remarks

Indeed, as the current project results showcase, CCIs' interventions within innovation discourse can be manifold. Based on the qualitative innovation cases (case study) from the tourism sector, this work reveals CCIs' intervention within the S3 policy nexus as well as sustainability and resilience considerations. Project results are disputed in detail in the Results chapter. For this purpose, and in order to not necessarily repeat that already mentioned, the results will be discussed here by mainly linking them to the applicable research streams regarding S3 policy and highlighting CCIs contribution.

This research work is the first attempt to integrate creative potential of CCIs into the S3 policy nexus beyond just linking up S3 and cultural heritage. In addition, the work takes rather an untapped research journey and aims at revealing CCIs' potential for streamlining and improving local and regional innovation capacities through S3 policy, mainly for innovation co-creation with CCIs in the EU tourism service sector. CCIs' brokerage is grasped in the micro-level S3 policy perspective, i.e., on the entrepreneurial/firm-level, as well as placed within the entire eco-system (meso-level) and regional/national system level (macro-level). Given the strong conceptual relatedness of CCIs and S3 (Entrepreneurial Discovery Process), CCIs can be integrated into S3 policy design and implementation as well as from the processual and institutional perspectives. Respectively, processual CCIs intervention is reflected through creative auditing and the adapted Creative Audit Tool for S3-driven innovation on the one hand, and a creative broker strengthening Quadruple helix actors' engagement, integration, co-creation, co-evolvement, integration and institu-

tionalization throughout the entire innovation or, in S3 terms, the self-discovery process. In doing this, this work yields trifold research contributions.

First, the present paper contributes to enhancing the empirical body on micro-level impacts on S3 policy. This is rooted in the exploration of innovation development and co-creation initiatives performed on the entrepreneurial/SME level in the particular region and under the scrutiny of the tourism sector, thus addressing micro-level gaps and exploring the perceptions of stakeholders from quadruple helix partnerships [122] (p. 2). Second, the research undertaken provides new pathways for perceiving and integrating sustainability and resilience dimensions into the S3 policy realm, specifically through CCIs' conceptual interlinkage with sustainability, capability and proximity approaches. The results showcase how CCIs can support entrepreneurial agents and other actors from the quadruple helix engaged in co-creation to build up the capacity, which is necessary for sustainable environmental, economic, social and institutional thinking and acting. Advocating a holistic paradigm, driven by human-centered motives, CCIs are able to broker and transfer ecological consciousness, resource efficiency, social inclusion and institutional stability. With the proposed Creative Audit Tool and with the counselling potential of creative brokers, innovation agents on the local and regional level are able also to build up and/or enhance preparatory, absorptive, recoverable and adaptive resilience capability, thus enabling them to better react to disruptive changes and to prepare for the upcoming future disruptions. With this contribution, the present work fits into the recently challenged oriented innovation policy, which is especially crucial in the face of the commenced twin transition and increase pace for transformation, as multi-scalar and inter-scalar coordination appears to be pivotal for managing regional sustainability transition [199] (p. 1). Thus, this research enhances the understanding and meaning of quadruple helix partnerships and their correlation with the sustainability efforts. In particular, the research positions CCIs within quadruple helix networks and claims that CCIs should be integrated far beyond just simply putting them into the "society" actors' basket, since cultural and creative dimensions do not target any particular helix, but rather the entire environment (ecosystem) itself [139]. For this, creative brokers could be introduced as a fifth dimension in a helix, connecting all concerned stakeholders, integrating, accumulating and transferring knowledge, mediating through conflict resolution and advocacy as well as becoming system changers through intervention and reaching out to all affected actors in the entire ecosystem and beyond this (intra-ecosystem co-existence). This would support early-stage research into N-helix emergence and its role in sustainable development expressed through technological, social and specialized innovations [200].

Bearing in mind the presented and discussed results, the authors claim having delivered both theoretical and managerial contributions. From the scholarly perspective, with the anticipated aim and exploration trajectory, this research contributes to the research stream of responsible research and innovation (RII), through accommodating alternative views and sources (creativity) to sustainable transformation, leading to an increased shared value within the process of innovation and quadruple helix interactions, by engaging, implementing, and delivering innovation, which is in line with sustainability principles. Thus, CCIs' intervention and contribution towards both innovation and institutional arrangements supports transformative innovation policy [201,202]. Moreover, addressing knowledge proximity and networks as well as innovation capacity and capabilities, this work enriches the resource-based view (RBV) and dynamic capabilities related firm level theories [203]. Finally, through local and regional dimension and exploration of S3 policy design and implementation, an overall contribution is made to the economic geography's research streams.

Managerial implications are mainly linked with entrepreneurial/firm competence portfolio enrichment. With this research, local and regional SMEs engaging in innovation development and realization processes are provided with the opportunity to utilize co-creation tools and exploit counselling and creative contribution of CCIs. These tools provide a holistic overview and make it possible in the micro-level system to intervene and connect

with the meso and macro levels in terms of the S3 domain. In addition, this contribution can serve as an inspirational or compassing tool for SMEs searching for new ways to overcome recession and depression, mainly in the tourism sector, as well as turning their touristic services into desirable and creative ones. As a result, this also contributes to growing practical policy contributions on the topic of creative tourism [204]. In addition, SMEs transfer their places of action and wellbeing into places of environmental, social and economic wellbeing, as well as promoting tourism based on creativity as a way to stabilize, recover and advance communities, through the coupling of dynamic creative relationships between people, places, institutions, ideas [205,206].

This research is the first attempt to bring in CCIs into the S3 policy dimension. Rationally, a series of prosperous future research avenues arises. Since this research is based on qualitative data, the next step is to multiply and validate the research contributions. In addition, the following research needs to bring in stronger CCIs' intervention and contribution to the 2021–2027 cohesion and regional policy period. This, however, can be completed upon the publishing of new regional innovation and smart specialization Strategies, which to a large extent are still under development. Practically, the authors recommend increasing cross-sectoral collaboration and the utilization of creative potential both on the micro- as well as the meso- and macro-level industry and policy levels. It is widely acknowledged that CCIs can boost regional development and better equip SMEs with tools and methods to react to future challenges as well as to implement ambitious environmental and social goals as set for 2050. In the short-term, 2030 should mark the next pivotal CCI evaluation period in terms of the implementation of the Territorial Agenda 2020 of the EU and achievements of the Sustainable Development Goals of the United Nations.

**Author Contributions:** C.M. made the overall conceptualization of the present research work by incorporating design and business policy perspectives of the succeeding actors. The outline, proposal of the theoretical and analytical phases belonged to the responsibility of C.M. The research design and choosing the research design, methods and data were coordinated, suggested and monitored by C.M. C.M. is responsible for S3-related policy results' analysis and evaluation, harmonization with the existing initiatives as well as formulation of conclusions and implications for research and practice. L.G. is responsible for integrating CTCC project perspective into the overall concept, including the conceptualization of the CCIs conceptual foundation and interlinkage with other theoretical concepts. Creative Audit Tool and creative broker concepts were adapted to the needs of the S3 policy nexus by L.G. and M.K. made her contribution through analysis and evaluation of Polish innovation cases and their comparison with overall research results. C.M. and L.G. shared responsibility over overall paper editing, its improvement and finalization. All authors have read and agreed to the published version of the manuscript.

**Funding:** This research was supported by the Interreg V A project "CTCC—Creative Traditional Companies Cooperation", part-financed by the South Baltic Program 2014–2020, respectively.

**Institutional Review Board Statement:** Not applicable.

**Informed Consent Statement:** Not applicable.

**Data Availability Statement:** The data presented in this study are partly openly available and can be retrieved via the project websites (https://movecreative.eu, accessed on 31 January 2022). Additional data of this study are available on request from the corresponding author.

**Acknowledgments:** We thank the innovation prototyping SMEs for disclosing their data for the project (research purposes) and engaging in the innovation journey with the open/unknown outcomes of innovation at the moment of engagement. We thank you for your trust and belief in the project.

**Conflicts of Interest:** The authors declare no conflict of interest. All project results are publicly available as a result of public funding of the CTCC project (European Regional Development Fund 2014–2020). Intellectual property rights for innovation prototypes' analysis, their evaluation, positioning within the scholarly literature and management practices remain in the ownership of the authors of this paper.

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
