# Peer review of "Creativity as a Key Constituent for Smart Specialization Strategies (S3), What Is in It for Peripheral Regions? Co-creating Sustainable and Resilient Tourism with Cultural and Creative Industries"

_sustainability, doi:10.3390/su14063469_

Round 1

Reviewer 1 Report

The structure and the readability of the paper could be improved, avoiding unnecessary repetitions and redundancy. The outline of the entire work, the research questions and the hypothesis could be better presented in the introduction.

Focus on partnerships interaction and collaboration governance is missing (even if it is mentioned on row no. 74 of the introduction "as emphasized in the previous paragraph") and necessary.

Clarity of the presentation of the case studies could be improved, highlighting the link with the research questions mentioned in the introduction and the related findings.

Author Response

Thank you for you remarks. Please find attached the response of the research team.

Reviewer 2 Report

Comments on ID1610052:

Creativity as Key Constituent for Smart Specialization. What’s 2 in it for Peripheral Regions? Co-Creating Sustainable and Resilient Tourism with Creative Industries

General comment:

This study discusses the Smart Specialization Strategies (S3) policy design and implementation on CCIs in region of South Baltic Sea Region (SBSR). The authors find that the SBSR becomes vulnerable to disruptions, e.g. caused by COVID-19, after focusing on specialization. This study contributes to expand literature regarding policy implementation in supporting innovation development on the peripheral localities. However, the article tries to convey lots of thinking around innovation, CCIs and sustainability, but lacks clear description of the content of S3 policy and the linkage between the implementation of S3, innovation and sustainability on surveyed cases.

Method and results:
This study conducts analysis on  sustainability and resilience of the 7 cases of CCIs in tourism sector located at SBSR. There are some questions regarding the methodology:

  • This article discusses “Innovation” a lot in section 2.1 and 2.2 and mentions S3 sporadically. Suggest making a more clear description in a separate paragraph regarding the policy content of S3.
  • Table 1 lists the “Shadowing, co-creation, expert interviews, participatory observation” as research methods while there  is “ expert interviews “applied mainly in this article.
  • Table 1 lists the “Qualitative + quantitative” as research data, but there seems no  quantitative data inside this article.
  • For Research techniques in  Table 1, “self-discovery” is an uncommon technique in research paper. Inductive analysis is much more appropriate.
  • Table 3 summarizes the case analysis and grading on “sustainability and resilience” by 4 regions. As shown in Table 2, cases in same district might focus differently as shown in case Maxim. For example, LT1, LT2, LT3 have different potentials on sustainability and resilience. Suggest adding a new table to analyze based on different dimensions (sustainability and resilience ). This will help to understand the policy implementation and its impact on sustainability of the 7 cases.
  • Table 3, the “NN” in the intervention means ?
  • Table 3 uses four plus “++++” to grade sustainability and resilience dimensions of each cases. The author describes the grading process and explain the missing of “plus” for some cases. However, what’s the grading content and criteria. There should be a grading criteria clearly described first then comes out a grading result, particular on missing items.
  • There is the same title as “Figure 1” for the first two figures.
  • Figure 1 introduces concept of “Creative Audit Tool and Creative Brokers”. How this tool impacts the 7 cases on innovation and sustainability?

Discussions:

(1) The article mentions: “This research work is the first attempt to integrate creative potential of CCIs into the policy nexus beyond just linking up S3 and cultural heritage.”  There is many descriptions regarding innovation concept in literature section while the linkage between innovation and the S3 policy implementation of the 7 cases is not much, at least not shown in the content of Table 1-3.

(2) The authors recommend “ to increase adventure of cross-sectoral collaboration and utilization of the creative potential both on micro- as well as macro-level industry and policy levels. It is widely acknowledged that CCIs can boost regional development and better equip with tools and methods to react to future challenges as well as implement ambitious environmental and social  goals as set for 2050.” How to obtain the relevant collaboration with different level institutions? If now the 7 cases show vulnerable to disruptions after the implementation of S3 policy, there is needs for the adjustment for the policy implementation?

Author Response

(The authors gave the same response as above.)

Round 2

Reviewer 2 Report

The authors have revised this manuscript according to comments. This article can be accepted.